# The Consumption of Beeswax Alcohol (BWA, Raydel^®^) Improved Zebrafish Motion and Swimming Endurance by Protecting the Brain and Liver from Oxidative Stress Induced by 24 Weeks of Supplementation with High-Cholesterol and D-Galactose Diets: A Comparative Analysis Between BWA and Coenzyme Q_10_

**DOI:** 10.3390/antiox13121488

**Published:** 2024-12-05

**Authors:** Kyung-Hyun Cho, Yunki Lee, Ashutosh Bahuguna, Sang Hyuk Lee, Chae-Eun Yang, Ji-Eun Kim, Hyo-Shin Kwon

**Affiliations:** Raydel Research Institute, Medical Innovation Complex, Daegu 41061, Republic of Korea

**Keywords:** beeswax alcohol (BWA), behavioral analysis, brain, coenzyme Q_10_ (CoQ_10_), D-galactose, senescence, swimming distance, myelin

## Abstract

The prolonged consumption of D-galactose (Gal) has been associated with severe damage in the liver and brain via exacerbation of oxidative stress, non-enzymatic glycation, and the aging process. The current study was initiated for a comparative assessment of beeswax alcohol (BWA, final 0.5% and 1.0% *w*/*w*) and coenzyme Q_10_ (CoQ_10_, final 0.5% and 1.0% *w*/*w*) against high-cholesterol (HC, final 4%, *w*/*w*) and -galactose (Gal, final 30%, *w*/*w*)-induced adverse events in zebrafish during 24 weeks of consumption. The survivability of zebrafish decreased to 82.1% due to HC+Gal exposure, but this was substantially improved (91.0%) with the consumption of 0.5% and 1.0% BWA. In contrast, no protective effect of CoQ_10_ consumption (1.0%) was observed on the survivability of zebrafish. Nevertheless, both BWA and CoQ_10_ displayed a significant (*p* < 0.001) preventive effect against HC+Gal-induced body weight enhancement. The HC+Gal-induced cognitive changes, marked by staggered and confused swimming behavior, and retarded swimming speed and motion patterns (restricted to the bottom of the tank), were efficiently restored by BWA. A significantly higher residence time in the upper half of the tank, 3.1-and 4.5-fold reduced latency time along with 3.5-fold and 4.1-fold higher swimming distance, was logged in the 0.5% and 1.0% BWA groups, respectively, than the zebrafish that consumed HC+Gal. In addition, BWA effectively enhanced plasma ferric ion reduction (FRA) and paraoxonase (PON) activity and alleviated the total cholesterol (TC), triglyceride (TG), and blood glucose levels disrupted by the consumption of HC+Gal. Also, the HC+Gal-alleviated plasma high-density lipoprotein-cholesterol (HDL-C) was 2.6-fold (*p* < 0.001) enhanced in the group that consumed 1.0% BWA, which was significantly 1.5-fold (*p* < 0.001) better than the effect of 1.0% CoQ_10_. Similarly, BWA displayed a superior impact over CoQ_10_ to mitigate HC+Gal-induced plasma AST and ALT levels, hepatic IL-6 production, generation of oxidized species, cellular senescence, and fatty liver changes. Moreover, BWA protects the brain against HC+Gal-induced oxidative stress, apoptosis, and myelin sheath degeneration. Conclusively, compared to CoQ_10_, BWA efficiently can the HC+Gal-impaired brain and liver functionality to subside and improves the dyslipidemia and cognitive behavior of zebrafish.

## 1. Introduction

The long-term consumption of D-galactose (Gal) has been widely acknowledged as a contributor to aging in major organs, including the liver, heart, brain, and skin [1,2,3,4], primarily through non-enzymatic glycation and elevated oxidative stress [5,6]. In a rodent model, Gal consumption reduced longevity by exacerbating cognitive dysfunction, neurodegeneration, and liver inflammation [7,8]. The pathogenesis of galactose-induced cataracts, vision loss, and blindness due to opacity of the lens is associated with the production and accumulation of reactive oxygen species (ROS), protein denaturation, and cellular damage [9,10]. In addition, Gal-induced aging is associated with dyslipidemia, high serum cholesterol, and triglycerides with low HDL-C, which are the hallmarks of metabolic aging and fatty liver disease [11]. Because high cholesterol (HC) also induces cellular senescence and aging via an increase in ROS production, the co-consumption of HC and Gal might exacerbate the pro-oxidative and pro-inflammatory cascade to cause aging [12,13].

Beeswax alcohol (BWA), a mixture of six long-chain aliphatic alcohols (tetracosanol, hexacosanol, octacosanol, triacontanol, dotriacontanol, and tetratriacontanol) obtained from beeswax (*Apis mellifera*) [14], has been well documented to ameliorate gastrointestinal disorders, peptic ulcers, and osteoarthritis via the protection of lipid and protein in the cell membrane with potent antioxidant and anti-inflammatory activity [15,16,17]. Due to its many health benefits, the Korean Ministry of Food and Drug Safety has approved BWA as a functional food ingredient to improve gastrointestinal and joint health [18].

CoQ_10_ is an intracellular antioxidant that prevents cellular senescence and dysfunction from oxidative stress to treat cardiomyopathy [19]. A meta-analysis of randomized controlled trials revealed that CoQ_10_ supplementations, 60–300 mg/day, among patients with coronary artery disease was also associated with an increase in serum HDL-C, superoxide dismutase (SOD), and catalase activity and decreased malondialdehyde levels [20]. A comparative analysis revealed that BWA exhibited a greater antioxidant effect than vitamin C [21] and coenzyme Q_10_ (CoQ_10_) [22] through its ability to inhibit LDL oxidation [21,22] and enhance HDL-associated paraoxonase (PON-1) activity [22].

Recently, we reported that 20 weeks of dietary supplementation of BWA effectively countered the high-cholesterol diet-induced dyslipidemia and protected the vital organs [23]. In a recent study, 12 weeks of BWA supplementation effectively mitigated HC (final 10%, *w*/*w*)+Gal (final 10%, *w*/*w*)-induced dyslipidemia, liver damage, inflammation, and impairment of kidney and reproductive organ function in zebrafish [24]. Moreover, the consumption of BWA ameliorated the dyslipidemia, production of ROS, and cellular senescence actuated by the consumption of HC+Gal. Despite the beneficial effect of BWA, the study [24] was limited only to the short-duration (12 weeks) consumption of HC+Gal with a relatively lower amount of Gal, which was noted to be insufficient in causing severe detrimental effects in the brain and cognitive changes. Given this, the present investigation aimed to compare the in vivo efficacy of BWA (final 0.5% and 1.0%, *w*/*w*) and CoQ_10_ (final 0.5% and 1.0%, *w*/*w*) against long-term (24 weeks) consumption of a high amount of Gal (final 30%, *w*/*w*) and cholesterol (final 4%, *w*/*w*), which induced severe detrimental effects in the brain, aging, and cognitive impairments in the zebrafish (*Danio rerio*).

## 2. Materials and Methods

### 2.1. Materials

The beeswax alcohol (BWA) was complementarily provided by Raydel^®^ Pty, Ltd. (Thornleigh, NSW, Australia). The BWA (>86% purity) was extracted from the block beeswax of honeybee (*Apis mellifera* mainly mellifera linage). The extracted BWA contained a mixture of six long-chain aliphatic alcohols (LCAA): C24, C26, C28, C30, C32, and C34. Detailed specifications of the BWA (batch#:330020123) and the certificate of analysis are listed in Appendix A. D-galactose (Gal, Cat # G5388), coenzyme Q_10_ (CoQ_10_, Cat # 303-98-0), and 5-bromo-4-chloro-3-indolyl β D-galactopyranoside (X-gal, Cat # B4252) were purchased from Sigma-Aldrich (St. Louis, MO, USA). All the other chemicals and reagents were of analytical grade and used as supplied. The myelin specific primary antibody was in-house produced in rabbits and was complementarily provided by Prof. Cheol-Hee Kim (Laboratory of Developmental Genetics, Department of Biology, Chungnam National University, Daejeon 34134, Republic of Korea). The secondary antibody Goat anti-rabbit IgG (AlexaFluor^TM^ 405, A-31556) was purchased from Thermo Fischer (Waltham, MA, USA).

### 2.2. Formulation of Different Diets

The normal tetrabit (ND), a regular diet of zebrafish, was mixed with cholesterol (final 4%, *w*/*w*) to make a high-cholesterol (HC) diet. The HC was blended with galactose (Gal, final 30%, *w*/*w*) to make an HC-supplemented galactose diet (HC+Gal). HC+Gal was individually amalgamated with BWA (final 0.5% or 1.0%, *w*/*w*) or CoQ_10_ (final 0.5% or 1.0%, *w*/*w*) to make four distinct HC+Gal formulations enriched with BWA and CoQ_10_ and abbreviated as HC+Gal+0.5% BWA or 1.0% BWA and HC+Gal+0.5% CoQ_10_ or 1.0% CoQ_10_, respectively. Table 1 depicts the specific amount of tetrabit, galactose, cholesterol, BWA, and CoQ_10_ used to prepare the different zebrafish diets.

The selection of 30% Gal was based on the preliminary screening study where Gal (10–50%) was supplemented in the ND and HC and fed separately to the zebrafish (*Danio rerio*). The study outcome revealed no substantial changes in the zebrafish mortality, weight, total cholesterol, glucose level, and cognitive behavior in the ND+Gal (10–50%) groups compared to the ND (control) group. However, in the HC+Gal groups, substantial adverse effects were observed with an increasing concentration of Gal (20–50%). Notably, severe zebrafish mortality was observed at HC+Gal > 30% (i.e., 40% and 50%). Therefore, 30% Gal with HC was selected as a relevant dietary composition to induce damaging effects on the zebrafish.

### 2.3. Zebrafish Husbandry and Consumption of Different Diets

Zebrafish (*Danio rerio*) AB strains were maintained at 28 °C water temperature in an alternative 14 h (light) and 10 h (dark) photoperiod following the standard guidelines of Animal Care Committee and Used approved by Raydel Research Institute (RRI, approval code RRI-23-007, approval date 27 July 2023).

Adult zebrafish (~18 weeks, n = 392) were randomly segregated and maintained with a normal diet (ND, n = 56) and a high-cholesterol diet (HC, n = 336). Following four weeks of consumption, the ND-fed zebrafish (Group I) were further maintained only on the ND diet for the next 24 weeks, while the HC-fed zebrafish were further segregated into five groups (n = 56/group). The zebrafish in Group II were maintained in the HC diet only, while the zebrafish in Groups III and IV were fed with HC+Gal+0.5% BWA and HC+Gal+1.0% BWA, respectively. The zebrafish in Groups V and VI were fed with HC+Gal+0.5% CoQ_10_ and HC+Gal+1.0% CoQ_10_, respectively. The experimental layout involves feeding zebrafish with different diets is depicted in Figure 1. The zebrafish in the different groups (I–VI) were fed the designated diet (10 mg/zebrafish) twice a day (morning and evening) until 24 weeks. The food consumption was measured by using the following formula: [(given food amount − remaining food amount)/given food amount] × 100. The feed efficiency was quantified by measuring the feed conversion ratio using the following equation: [weight gain (g)/feed consumed (g)] × 100. The feed efficiency was calculated at 24 weeks of consumption of the respective diets.

The mortality of zebrafish across all the groups was periodically measured until 24 weeks of the experimental period. For the body weight analysis, zebrafish were anaesthetized by submerging them in a 0.1% solution of 2-phenoxyethanol. The anaesthetized zebrafish were surface-dried using filter paper, and body weight was analyzed using an electronic balance (Ohaus, Parsippany-Troy Hills, NJ, USA).

### 2.4. Behavioral Analysis of Zebrafish

The behavioral analysis of zebrafish across all the groups was analyzed using an open field tank following video recording. For the behavioral analysis, the filled water area was divided into two halves using horizontal lines in the tank. The divided water section was named the lower and upper half [as depicted in Appendix A]. The zebrafish from the different groups were transferred into the respective tanks (divided into two halves). Following 1 h of acclimatization, the zebrafish’s swimming movement was recorded (1 min). The video of swimming movement was analyzed using “Tracker video analysis and modelling software” (https://physlets.org/tracker, assessed on 15 August 2024) to quantify total swimming distance, swimming trajectory, swimming speed, total time spent in the tank’s upper and lower halves, and latency time.

### 2.5. Collection of Blood and Organs

After 24 weeks of consumption of different diets, the zebrafish from the respective groups were sacrificed using hypothermic shock [25]. Immediately, blood was collected in ethylenediamine-tetraacetic acid (EDTA) rinsed tubes and, further, processed for the quantification of total cholesterol (TC), triglycerides (TGs), high-density lipoprotein-cholesterol (HDL-C), and the hepatic function biomarkers aspartate aminotransferase (AST) and alanine aminotransferase (ALT) using commercial kits following the recommended methodology suggested by the manufacturer. A detailed procedure for the blood analysis is provided in Appendix A. The organs (liver and brain) from the zebrafish were surgically removed and preserved in 10% formalin for further histological analyses.

### 2.6. Estimation of Blood Glucose, Ferric Ion Reduction (FRA), and Paraoxonase (PON) Activity

Blood from the different groups was analyzed using an electronic blood glucose meter (AccuCheck, Roche, Basel, Switzerland) to estimate glucose, and the results are depicted as glucose mg/dL. The ferric ion reduction ability (FRA) of plasma was examined using the method described earlier [26]. Briefly, 20 μL of plasma (0.1 mg equivalent protein) was mixed with 180 μL of FRA reagent [prepared by blending 10 mM 2,4,6-tripridyl-S triazin (1.25 mL) and 20 mM ferric chloride dissolved in 10 mL of 0.2 M acetate buffer (pH 3.6)]. The content was incubated for 60 min at room temperature, followed by an absorbance assessment at 593 nm.

Plasma paraoxonase (PON) activity was determined by the earlier described spectroscopic method with slight modification [26]. In brief, a 40 μL plasma (0.1 mg equivalent protein) sample was mixed with 160 μL of 0.55 M paraoxon-ethyl [dissolved in 90 mM Tris-HCl, 3.6 mM NaCl, and 90 mM CaCl_2_]. After 60 min of incubation at 25 °C, absorbance at 415 nm was recorded, and PON activity was quantified as μU/L/min using a molar absorbance coefficient (ε) of 17,000 M^−1^cm^−1^ for the formed product *p*-nitrophenol.

### 2.7. Histological Analysis

The tissue sections (liver and brain) were individually dehydrated using ethanol and then embedded in a paraffin block. A thin tissue section (7 μm thick) was obtained from the cryo-sectioning using a microtome (Leica CM 1510S, Nussloch, Germany). The sliced tissue section was processed for hematoxylin and eosin (H&E) staining [27] to examine the changes in the liver and brain.

The senescence in the tissue sections of the liver and brain was analyzed with an SA-β-gal assay [28]. Briefly, the tissue section (7 μm) was fixed with 4% paraformaldehyde, followed by washing with PBS twice and the subsequent addition of X-gal (250 μL, 1 mg/mL). Following 16 h of incubation at room temperature in the dark, the stained section was washed three times with PBS and visualized under a microscope. Oil red O (ORO) staining [25] was performed to observe the fatty liver changes in the liver tissue. In brief, 250 μL of ORO (3 mg/mL) was poured over the tissue section (7 μm). Following 10 min of incubation, the section was washed with water and visualized under a microscope.

### 2.8. Immunohistology (IHC)

For the detection of hepatic interleukin (IL)-6, an immunohistochemical analysis was performed using the target-specific primary antibodies [29]. For the IL-6 detection, the hepatic section (7 μm) was incubated overnight with IL-6 specific antibody (ab9324, Abcam, London, UK) following IHC development with a secondary antibody containing the EnVision + System-HRP polymer kit (Code K4001, Dako, Glostrup, Denmark).

The myelin sheath in the brain was detected by immunohistochemical analysis using a myelin-specific antibody. For the myelin sheath detection, the zebrafish myelin-specific primary antibody from rabbit (200× diluted) was spread over the brain tissue section (7 μm thickness). Following overnight incubation at 4 °C, the tissue section was washed three times with PBS, followed by the subsequent addition of goat anti-rabbit IgG (AlexaFluor^TM^ 405, A-31556, Thermo Fischer). After 4 h of incubation in the dark at 4 °C, the tissue section was washed three times with PBS and visualized under a fluorescent microscope at the excitation (Ex) and emission (Em) wavelengths of 401 and 422 nm, respectively.

### 2.9. Fluorescent Staining for Reactive Oxygen Species (ROS) Production and Apoptosis

Dihydroethidium (DHE) [30] and acridine orange (AO) [31] fluorescent staining was performed to detect ROS and the extent of apoptosis. In brief, 250 μL of DHE (30 mM) and AO (30 mg/mL) was poured on the tissue section (7 μm) and incubated in the dark for 30 min. The stained section was rinsed three times with PBS and visualized under a fluorescent microscope at 585 nm (Ex)/615 nm (Em) and 505 nm (Ex)/535 nm (Em) for the detection of DHE and AO fluorescent stained areas, respectively.

### 2.10. Statistical Analysis

The results in the graphs are represented as mean ± SEM obtained from the triplicate experiments. The statistical significance (*p* < 0.05) between the groups was established using a one-way analysis of variance (ANOVA) following Tukey’s post hoc analysis, while the pairwise assessment was examined by T-test using the Statistical Package for Social Science software program (version 29.0, Chicago, IL, USA).

## 3. Results

### 3.1. Change in Survivability and Body Weight During Galactose Consumption

The Kaplan–Meier survival analysis, followed by a log-rank test, revealed a significant difference (log-rank: *χ*^2^ = 13.5, *p* = 0.035) in zebrafish solvability among the different groups. As represented in Figure 2A, the highest survival probability (1.0) was maintained in the ND control group throughout the 24 weeks of consumption. In contrast, the HC group experienced a substantial drop in survival probability (0.88), which declined to 0.81 in the HC+Gal group over the 24 weeks of consumption. The HC+Gal-induced mortality of zebrafish was substantially prevented by the consumption of BWA, as reflected by the 0.89 survival probability in the group that consumed BWA 0.5% and 1.0% for 24 weeks. In contrast to BWA, the CoQ_10_ consumption showed a lower effect on HC+Gal-induced zebrafish mortality, even more in the group that consumed 1.0% CoQ_10_; 0.81 survival probability was observed, which is analogous to the survival probability observed in the HC+Gal group.

Among all the groups, ~95–100% food consumption was noticed within 30 min of feed supply (Table 2). The feed efficiency ratio among the groups was in the range of 6.3–14.6%. The highest feed efficiency was seen in the group that consumed HC+Gal (14.6%), while the least was in the group that consumed ND (6.3%), followed by the 1.0% BWA (6.8%), 0.5% BWA (7.4%), 0.5% CoQ_10_ (8.3%), and 1.0% CoQ_10_ (8.6%) groups (Table 2).

As illustrated in Figure 2B, after 24 weeks, the HC+Gal group exhibited the most pronounced body weight gain (870.6 ± 51.7 mg), approximately 2.3-fold higher from week 0 (377.4 ± 12.7 mg), followed by the HC alone group (731.5 ± 16.4 mg), which displayed around a 1.8-fold increase from the baseline (week 0, 398.6 ± 20.1 mg). The HC+Gal-mediated BW enhancement is significantly (*p* < 0.001) controlled by the consumption of BWA and CoQ_10_.

### 3.2. Swimming Behavioral Analysis of Zebrafish

A substantial variation in swimming behavior concerning the swimming distance and swimming pattern was noticed across the different groups (Appendix A). The heat map of the swimming trajectories (obtained at 22 weeks of consumption) showed a regular swimming pattern of zebrafish (free movement across the tank) in the ND and BWA groups (Figure 3A). Similarly, the zebrafish in the HC and CoQ_10_ groups displayed free-swimming movements; however, the occasional thigmotaxic behavior (time spent at the bottom and side of the tank) was also noticed. In contrast, the HC+Gal group exhibited a marked cessation of swimming behavior, characterized by pronounced bottom-dwelling and thigmotaxic tendencies.

The swimming behavioral studies at 24 weeks of consumption revealed a free exploration of zebrafish movement spanning in the horizontal and vertical directions across the tank (Figure 3B). A noticeable maximum swimming time in the upper half (65.7%), the least latency time (3.1 s, towards the top phase), and the maximum swimming distance (4.3 m) was noticed in the zebrafish of the ND group (Figure 3C–F). A severe burst with clumsy swimming that is highly restricted to the bottom side was noticed in the HC+Gal group. Contrary to the ND group, a significant 3.2-fold delayed latency time, 59-fold reduced swimming time in the upper half, and 4.6-fold reduced total swimming distance were noticed in the HC+Gal group. The HC+Gal-induced severity was effectively curtailed by the consumption of BWA at both the tested concentrations (0.5 and 1.0%). As seen in Figure 3B–D, free-swimming movement across the tank with a significant 3.0-fold (*p* < 0.05) and 4.5-fold (*p* < 0.01) better latency time was observed in the 0.5% and 1.0% BWA groups compared to the HC+Gal group. Likewise, the stay-up time in the top half was significantly, 33.9-fold (*p* < 0.001) and 42.6-fold (*p* < 0.001), higher, and 3.6-fold (*p* < 0.001) and 4.2-fold (*p* < 0.001) higher total swimming distances were observed in the 0.5% and 1.0% BWA groups than the HC+Gal group (Figure 3E). The consumption of CoQ_10_ also displayed a substantial preventive effect against the HC+Gal-induced swimming impairment of zebrafish. However, when compared with BWA, precisely at 1.0%, a significant 1.2-fold (*p* < 0.01) and 1.4-fold (*p* < 0.001) lower stay time in the top half and total swimming distance, respectively, was noticed in the 1.0% CoQ_10_ group.

### 3.3. Lipid Profile, Glucose Level, and Antioxidant Status of Blood

As shown in Figure 4A,B, the ND group showed the lowest plasma TC and TG levels, while the HC group showed 1.7-fold and 2.0-fold elevations in TC and TG levels to those of the ND group. Furthermore, the HC+Gal group showed the highest TC and TG levels, which were significantly, 2.0- (*p* < 0.001) and 2.5-fold (*p* < 0.001), higher than in the ND group and 1.2- (*p* < 0.001) and 1.3-fold (*p* < 0.001) higher than in the HC group. However, the co-consumption of BWA resulted in a remarkable decrease in TC and TG levels; the 1.0% BWA group showed the lowest levels of TC and TG, i.e., 50% and 54% less than the HC+Gal group. Similarly, the CoQ_10_ displayed 31–35% less TC and 18–30% lower TG levels than the HC+Gal group.

As shown in Figure 4C, the HC+Gal group showed the lowest HDL-C level, which was 49% lower than the ND group and 22% lower than the HC-consumed group, respectively. The supplementation of both BWA and CoQ_10_ displayed a substantial effect on the elevation of HDL-C disturbed by the consumption of HC+Gal. However, the BWA at 1.0% consumption displayed the most promising results by elevating the HDL-C level by 1.8-fold (*p* < 0.001) and 1.4-fold (*p* < 0.001) more than those of 0.5% and 1.0% CoQ_10_, respectively.

A significant elevation in the blood glucose level was observed in the HC+Gal group (57.0 mg/dL) than the ND (30.5 mg/dL) and HC (47.5 mg/dL) groups (Figure 4D). Consumption of both BWA and CoQ_10_ effectively reduced the HC+Gal-elevated blood glucose level, as reflected by the 42.0 mg/dL and 50.5 mg/dL blood glucose in the 1.0% BWA and 1.0% CoQ_10_ groups.

The plasma antioxidant status of zebrafish among the different groups was determined by an FRA assay (Figure 4E). The HC+Gal group showed a lower (636.4 μM, *p* < 0.01) FRA value than the ND group (888.6 μM), which was significantly elevated by 42% (*p* < 0.01) and 53% (*p* < 0.01) in response to the 0.5% and 1.0% BWA consumption groups, respectively. Similarly, a 34.4% (*p* < 0.05) higher FRA value was noticed in the 1.0% CoQ_10_ group than in the HC+Gal group.

A significantly compromised plasma PON activity (7.5 μU/L/min) was noticed in the HC+Gal group than the PON activity (17.1 μU/L/min) observed in the ND group (Figure 4F). The consumption of BWA displayed a dose-dependent effect and elevated PON activity by 2.0-fold (*p* < 0.01) and 3.0-fold (*p* < 0.001) in response to 0.5% and 1.0% BWA than the HC+Gal group. No significant effect on the elevation of PON activity was observed in response to the consumption of CoQ_10_.

### 3.4. Morphological Changes in Brain and Liver

Figure 5A–C depicts the morphology of the whole zebrafish, brain, and liver after 24 weeks of consumption of different diets. A negative effect of HC was noticed on the brain/body weight that became worse in the presence of galactose (HC+Gal), reflected by an 8% lower brain/bodyweight in the HC+Gal group than the HC group (Figure 5B,D). Notably, the most substantial protective effect was observed in the 1.0% BWA group, which displayed a significant elevation, 23%, 9%, and 12%, in the brain/body weight compared to the HC+Gal, and 0.5% and 1.0% CoQ_10_ groups, respectively. Besides the brain/body weight, the critical morphology analysis revealed that BWA consumption improved brain size and prevented shrinkage caused by HC+Gal, particularly in the cerebellum (marked by the red arrow).

As shown in Figure 5C,E, HC+Gal consumption displayed a substantial elevation in liver size and enhancement of liver/body weight, which was ~14% (*p* < 0.001) higher than the liver/body weight of the ND group. The co-consumption of BWA at both 0.5% and 1.0% effectively prevented HC+Gal-provoked hepatomegaly and decreased the liver/body weight by 25% (*p* < 0.01) and 32% (*p* < 0.001), respectively. Likewise, 0.5% and 1.0% CoQ_10_ also substantially prevented HC+Gal-induced hepatomegaly, evidenced by the 23% and 25% alleviated liver/body weight, respectively, compared to the HC+Gal group.

### 3.5. Histological Analysis of Brain Tissue

The histology analysis examined by H&E staining revealed a normal brain morphology with well-organized optic tectum (TeO) without hyperemia, with the least vacuolation, and with mononuclear cells with a clear zone in the ND group (Figure 6A,B). As opposed to the ND group, the HC+Gal-consumed group displayed an elevation in vacuolation (indicated by the blue arrow) and number of mononuclear cells with a clear zone (indicated by the red arrow) in the TeO and periventricular grey zone (PGZ) of the brain with the occasional hyperemia (indicated by the green arrow). The consumption of BWA prevented the HC+Gal-induced adverse changes. Precisely, BWA at 1.0% effectively reversed the HC+Gal-induced vacuolation and formation of mononuclear cells with a clear zone. Unlike BWA, CoQ_10_ at both the tested concentrations displayed less effect in managing the HC+Gal-induced impairment in the TeO region of brain tissue.

The immunohistochemical (IHC) staining revealed a ~15% lower myelin-specific fluorescent intensity in the HC-consumed group than in the ND group (Figure 6C,D). Particularly, substantial demyelination was noticed around the valvular cerebella (Val) (indicated by a brown arrow) and in the lateral recess of the diencephalic ventricle (LR) region of the brain section (indicated by a yellow arrow). The consumption of HC+Gal displayed a severe decline in myelin-specific fluorescent intensity, which was substantially, 2.4-fold (*p* < 0.001) and 2.0-fold (*p* < 0.01), lower than the fluorescent intensity observed in the control and HC groups, respectively, suggesting an adverse effect of HC+Gal on the myelin sheath. BWA consumption significantly prevents HC+Gal-induced demyelination, reflected by ~2.2-fold (*p* < 0.01) and 2.0-fold (*p* < 0.01) higher myelin-specific fluorescent intensity in the 0.5% and 1.0% BWA group than the fluorescent intensity observed in the HC+Gal group. Contrary to BWA, CoQ_10_ consumption displayed a non-significant effect in mitigating HC+Gal-provoked demyelination.

### 3.6. Reactive Oxygen Species (ROS) Production, Apoptosis, and Senescence in Brain

As shown in Figure 7, the ND-alone group showed the least red fluorescence (ROS production), green fluorescence (apoptosis), and SA-β-gal-stained areas (senescence), while the HC+Gal group showed the most ROS production, apoptosis, and senescence. In the HC+Gal group, prominent red and green fluorescence was detected across the PGZ and TeO (as indicated by the yellow arrow) that mainly concentrated adjacent to the valvular cerebella (Val) (as indicated by the white arrow) (Figure 7A,B). The most vigorous blue intensity (interchanged with red), corresponding to SA-β-gal-positive cells, were detected around the vascular lacuna of area postrema (Vas) and below the tectal ventricle (TeV) region (as indicated by the red arrow) (Figure 7D–F). The 0.5% and 1.0% BWA consumption groups displayed substantial, 1.5-fold (*p* < 0.001) and 1.6-fold (*p* < 0.001), reductions in DHE and 1.8-fold (*p* < 0.001) and 2.0-fold (*p* < 0.001) decreased AO fluorescent intensity compared to the respective fluorescent intensities that appeared in the HC+Gal group (Figure 7G,H). Similarly, the 0.5% and 1.0% BWA consumption group efficiently prevented the HC+Gal-induced cellular senescence manifested by a significantly, 3.3-fold (*p* < 0.001) and 4.6-fold (*p* < 0.001), reduced SA-β-gal-stained area, respectively, than the HC+Gal group (Figure 7I). On the other hand, no beneficial effect of CoQ_10_ consumption (at 0.5% and 1.0%) was observed in curtailing HC+Gal-induced ROS production. Likewise, CoQ_10_ consumption at 0.5% was found ineffective in preventing HC+Gal-induced apoptosis; however, a significant anti-apoptotic effect reflected by a 1.4-fold (*p* < 0.01)-reduced AO fluorescent intensity compared to the HC+Gal group was noticed in response to 1.0% CoQ_10_ consumption. Interestingly, CoQ_10_ at the tested concentrations of 0.5% and 1.0% effectively prevents HC+Gal-induced cellular senescence (Figure 7I); however, the effect is inferior (*p* < 0.05) compared to the 1.0% BWA group.

### 3.7. H&E Staining and Interleukin (IL)-6 Production in Liver Tissue

As shown in Figure 8A, H&E staining revealed that the HC+Gal group showed the highest neutrophil infiltration and H&E-stained area in the hepatic tissue. Co-consumption of BWA at 0.5% and 1.0% efficiently lowered the H&E-stained area around 46% (*p* < 0.001) and 65% (*p* < 0.001), and neutrophil counts by 60% (*p* < 0.01) and 64% (*p* < 0.01), respectively, compared to the HC+Gal group (Figure 8A,E). Contrary to BWA, co-consumption of CoQ_10_ at both the tested concentrations (0.5% and 1.0%) did not have a preventive effect against HC+Gal-induced hepatic damage. Notably, lipid droplets (indicated by the blue arrow) were highly prevalent in the HC+Gal group, which was substantially reduced with the consumption of 1.0% BWA.

Immunohistochemical (IHC) staining revealed a 2.6-fold higher IL-6 production in the HC-consumed group than in the control ND group (Figure 8C–F). The production of IL-6 induced by HC was significantly elevated by 1.5-fold (*p* < 0.001) when combined with galactose (HC+Gal), highlighting the detrimental role of galactose in exacerbating HC-induced inflammation (Figure 8F). The HC+Gal-induced IL-6 production was significantly, 2.1-fold (*p* < 0.001) and 2.9-fold (*p* < 0.001), reduced in the 0.5% and 1.0% BWA groups. Similarly, a 1.4-fold (*p* < 0.001) and 1.2-fold (*p* < 0.001) lower IL-6 production was observed in the 0.5% and 1.0% CoQ_10_ groups compared to the HC+Gal group. Notably, 1.0% BWA demonstrated significantly greater efficacy, inhibiting IL-6 production, reflected by the ~2.0-fold (*p* < 0.001) lower IL-6-stained area than the 0.5% and 1.0% CoQ_10_ groups, underscoring the superior activity of BWA over CoQ_10_.

### 3.8. Fatty Liver Change, ROS Production, Apoptosis, and Cellular Senescence

The HC+Gal group exhibited the most pronounced increase in dark red intensity from oil red O-staining, DHE, and AO fluorescence, which were 19-fold (*p* < 0.001), 4.4-fold (*p* < 0.001), and 5.8-fold (*p* < 0.001) higher, respectively, compared to the ND-alone group (Figure 9). The 1.0% BWA group displayed remarkable decreases of 16.6-fold (*p* < 0.001), 3.5-fold (*p* < 0.001), and 4.0-fold (*p* < 0.001) in the fatty liver changes (Figure 9A,E), ROS production (Figure 9B,F), and apoptosis (Figure 9C,G) actuated by the consumption of HC+Gal. In comparison, the CoQ_10_ groups did not show notable decreases in ROS production and apoptosis in both consumed doses of 0.5% and 1.0%. Contrary to ROS and apoptosis, the consumption of 0.5% and 1.0% CoQ_10_ effectively reverted the HC+Gal-induced fatty liver changes, reflected by 4-fold (*p* < 0.001) and 2.9-fold (*p* < 0.001) reductions in ORO-stained areas, respectively.

The SA-β-gal staining revealed higher senescence in the HC+Gal group that accounts for 64.8% SA-β-gal-positive cells, which was significantly 10-fold (*p* < 0.001) and 1.8-fold (*p* < 0.001) higher than the SA-β-gal-positive cells that occurred in the ND and HC-alone groups, respectively (Figure 9D,H). HC+Gal-induced cellular senescence was significantly prevented by the consumption of BWA, as reflected by the 5.5-fold (*p* < 0.001) and 6.1-fold (*p* < 0.001) lower SA-β-gal-positive cells in the hepatic section from the 0.5% and 1.0% BWA groups, respectively, compared to the HC+Gal group. Unlike BWA, the consumption of 1.0% CoQ_10_ displayed a non-significant effect in preventing HC+Gal-induced cellular senescence.

### 3.9. Plasma Parameters of Hepatic Damage

As shown in Figure 10, the plasma hepatic function biomarkers AST and ALT levels were significantly, 40% (*p* < 0.001) and 34% (*p* < 0.001), higher in the HC group compared to the ND group. The co-consumption of galactose with HC (HC+Gal) substantially elevated the HC-induced AST and ALT levels by ~10% (*p* < 0.01). The intake of both BWA and CoQ_10_ was found to be efficient in preventing HC+Gal-induced AST and ALT levels. Significantly, 25% (*p* < 0.01) and 30% (*p* < 0.01), lower AST (Figure 10A) and ALT (Figure 10B) levels were noticed in the 1.0% BWA groups than in the HC+Gal group, which was significantly, 12% (*p* < 0.01) and 21% (*p* < 0.001), better than the effect exerted by 1.0% CoQ_10_.

## 4. Discussion

Various animal studies describe the detrimental effects of high cholesterol consumption [32,33,34]. Similarly, the consumption of high amounts of sugar associated with obesity, type 2 diabetes, metabolic syndrome, and cardiovascular diseases has been documented [35]. This study revealed that the high consumption of Gal (40%) can cross the placenta and impair brain development of the fetus [36]. The excessive intake of reducing sugars also causes the non-enzymatic glycation of protein that successively produces distinct toxicogenic advanced glycation end products (AGEs) [37]. The high intake of both cholesterol and carbohydrates serves as an excellent dietary model for causing obesity, hepatic damage, and insulin resistance. It also resembles the pathology and progression of disease similar to humans [38].

The outcome highlights the higher zebrafish mortality and body weight gain in the HC+Gal group, compared to the HC group, testifying to the additive effect of Gal on the augmentation of HC-provoked mortality and body weight gain. HC+Gal-induced zebrafish mortality and body weight gain were substantially prevented by the consumption of BWA, which was substantially better than the effect exerted by CoQ_10_, attesting to the higher efficacy of BWA over CoQ_10_ in mitigating HC+Gal-induced mortality and obesity.

The adverse effect of high Gal on cellular antioxidants (SOD, catalase, and GSH-Px) [39], generation of free radicals, and induction of oxidative stress [40], which leads to cognitive disorder and brain aging has been established [40,41,42,43]. Consistently, we have also noticed a substantial adverse effect of HC+Gal on cognitive impairment and brain damage that was substantially restored by the consumption of BWA and CoQ_10_. However, compared to CoQ_10_, a much higher effect of BWA was observed at both the tested concentrations (0.5 and 1.0%). The higher impact of BWA over CoQ_10_ might be attributed to its substantially better antioxidant activity than CoQ_10_ [22], which also modulates the cellular antioxidants SOD, catalase, and GSH-Px [44,45].

Gal accumulation in the brain provokes glycation, leads to the generation of AGEs [40,46], and causes brain damage and cognitive change [47,48]. Therefore, a substance that can inhibit AGE formation substantially affects brain health and cognitive changes. The effect of BWA on preventing protein glycation has been recognized [21,22]; moreover, BWA, compared to CoQ_10_, displayed a much more superior activity that protects zebrafish embryos against carboxymethyllysine (CML) (a common AGE)-induced toxicity [22], underscoring the higher efficacy of BWA over CoQ_10_ in meeting CML-induced challenges.

The adverse effect of Gal has also been noticed to disrupt myelin [49], which is an important multicellular membrane of the central nervous system (CNS) and associated with cognitive processes associated with the brain [50]. The consumption of BWA significantly mitigated demyelination in response to HC+Gal. We speculate that the substantially high myelin content in the BWA group is due to its notable inhibitory impact on cellular ROS, which has been recognized among the main culprits that provoke demyelination [51] and the subsequent brain aging and cognitive changes [52].

Also, the findings of plasma FRA and PON activity support the antioxidant nature of BWA and align with earlier studies documenting the impact of BWA on the plasma oxidative variables and the modulatory effect on cellular antioxidants [53,54]. The significant influence of BWA in enhancing the activity of HDL-associated antioxidant PON underscores its role in improving HDL functionality, a critical particle with diverse beneficial properties. The results are aligned with previous in vitro findings, where BWA was documented to augment PON-1 activity, while no effect was observed for CoQ_10_ [22], strengthening the superior antioxidant modulatory effect of BWA over CoQ_10_.

Similarly, a higher potency of BWA than CoQ_10_ was observed in countering HC+Gal-induced dyslipidemia. The impactful role of CoQ_10_ against dyslipidemia has been described in animal and human studies [55]. On the contrary, the effect of BWA on dyslipidemia has been described in a limited number of studies. In one of our earlier studies, BWA effectively countered dyslipidemia induced by a high-cholesterol diet [23]. The high prevalence of long chain aliphatic alcohol (LCAA) like hexacosanol, triacontanol, and octacosanol in BWA are the key contributors responsible for the protective effect of BWA against HC+Gal-induced dyslipidemia. The assumption is based on earlier studies documenting the influence of hexacosanol, triacontanol, and octacosanol on cholesterol biosynthesis [56,57,58,59]. The pivotal role of hexacosanol in activating the AMP-activated protein kinase (AMPK) and subsequent inhibition of 3-hydroxy-methylglutaryl-coenzyme A reductase (HMG-CoA), a rate-limiting enzyme for cholesterol biosynthesis, has been noticed as the key event in managing cholesterol biosynthesis [57]. Moreover, a substantial modulatory effect of hexacosanol on the nuclear translocation of sterol regulatory element binding protein-2 (SREBP-2), which is an important transcriptional regulatory molecule for the expression of HMG-CoA, has been recognized as an essential event to regulate cholesterol biosynthesis [57]. A similar effect of octacosanol has been documented to inhibit cholesterol biosynthesis [58,59].

The role of galactose, high-fat diets, and oxidative stress in elevating blood glucose levels has been well documented, mainly due to their association with insulin resistance or impaired insulin secretion [60,61]. Herein, the HC+Gal group exhibited elevated blood glucose levels, which were significantly mitigated by supplementation with BWA. This effect of BWA can be attributed to its potent antioxidant properties, which help to counteract oxidative stress elevated by HC+Gal. Additionally, numerous studies have highlighted the ability of BWA to ameliorate insulin resistance [62], which consequently impacts blood glucose levels. These findings align with previous research demonstrating policosanol’s blood glucose-lowering effect in zebrafish and human subjects [63].

BWA also displayed a substantial protective effect against HC+Gal-induced hepatic damage that is substantially better than the effect of CoQ_10_. The results are aligned with those in an earlier published article, which suggests that the 6-month consumption of BWA cures fatty liver changes and insulin resistance in patients with non-alcoholic fatty liver disease (NAFLD) [62]. The better effect of BWA than CoQ_10_ can be highlighted by the higher antioxidant potential of BWA than CoQ_10_ [22], which substantially reduces oxidative stress in various organs [23]. Consistent with the hepatic histology results, significantly lower plasma hepatic function biomarker (AST and ALT) levels were observed in the BWA than CoQ_10_ group. The reduced hepatic function biomarkers (AST and ALT) in BWA can be linked with its effect on diminished TC and heightened HDL-C levels, as a negative correlation has been documented between the hepatic function biomarkers, and elevated HDL-C [64] and diminished TC [65]. The effect of hexacosanol (a BWA component) on the induction of autophagy [57] may also contribute to preventing liver damage, as autophagy has been recognized to modulate hepatic lipid accumulation and fatty liver changes [57,66].

The higher IL-6 levels in the HC+Gal group can be linked with heightened ROS production, as different studies established the inductive effect of oxidative stress on the inflammatory pathway [67]. Similar to the present study’s outcomes, many earlier studies depicted BWA’s role in preventing inflammatory disorders in mice [68] and humans [17], which supports the current findings. The lower extent of apoptosis in response to BWA than that of CoQ_10_ is due to the higher antioxidant activity of BWA, which effectively limits HC+Gal-induced ROS generation and leads to the inhibition of oxidative stress, which has been accredited as a main provocative agent in inducing apoptosis [69]. Likewise, oxidative stress has been recognized as the main provocatory of cellular senescence [70,71]. Compared to CoQ_10_, BWA demonstrates a higher impact on mitigating HC+Gal-induced oxidative stress, resulting in a more pronounced reduction in cellular senescence. As summarized in Figure 11, the antioxidant and anti-inflammatory nature of BWA is a key contributor towards its diverse functionality in preventing HC+Gal-induced senescence, oxidative stress, fatty liver changes, and brain impairment.

As BWA showed substantial hepatoprotective activity and restored the fatty liver changes induced by the consumption of HC+Gal, it has a positive impact on brain health and cognitive changes. This notion is supported by earlier studies documenting the liver–brain axis [72]. The hepatic encephalopathy caused by liver dysfunction leads to the accumulation of toxic substances in the brain and is characterized by cognitive changes [73]. Many reports documented the association of NAFLD with structural changes in the brain and enhanced risk of dementia [74]. Even an excessive accumulation of fat in the liver alters liver functionality and changes the composition of fat-derived molecules in the blood, which subsequently triggers inflammation in the blood–brain barrier and provokes the entry and accumulation of toxic substances and inflammatory cells into the brain [74], leading to several detrimental effects including cognitive changes.

## 5. Conclusions

This 24-week consumption study displayed the remarkable functionality of BWA in preventing HC+Gal-induced adverse events in the blood, brain, and liver of zebrafish. Compared to CoQ_10_, BWA displayed a much higher effect in restoring HC+Gal-induced dyslipidemia and altered hepatic function biomarkers, blood glucose levels, and plasma FRA and PON activities. Also, a substantial effect of BWA was noticed in preventing myelin sheath degeneration in the brain and in restoring swimming behavior in zebrafish altered by the consumption of HC+Gal. The results outlined the functional superiority of BWA over C_O_Q_10_ and endorsed its dietary supplementation to counter high-cholesterol and galactose-induced adverse events and the restoration of cognitive changes.

## Figures and Tables

**Figure 1 antioxidants-13-01488-f001:**
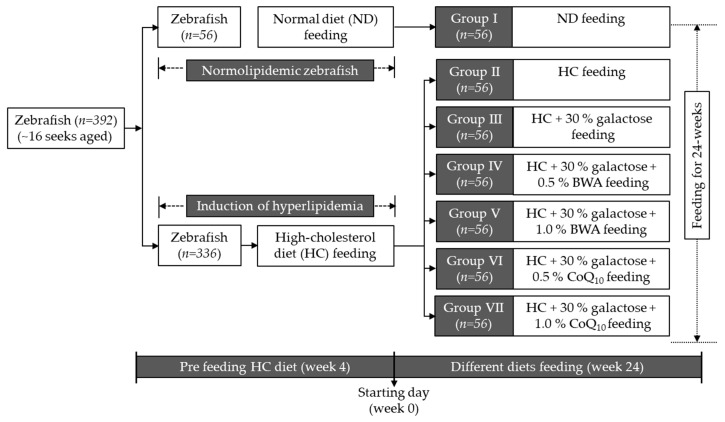
The experimental layout involves feeding zebrafish with different diets. The abbreviations: ND represent normal diet; HC represents high-cholesterol diet (final 4%, *wt*/*wt*); and BWA and CoQ_10_ represent beeswax alcohol and coenzyme Q_10_, respectively.

**Figure 2 antioxidants-13-01488-f002:**
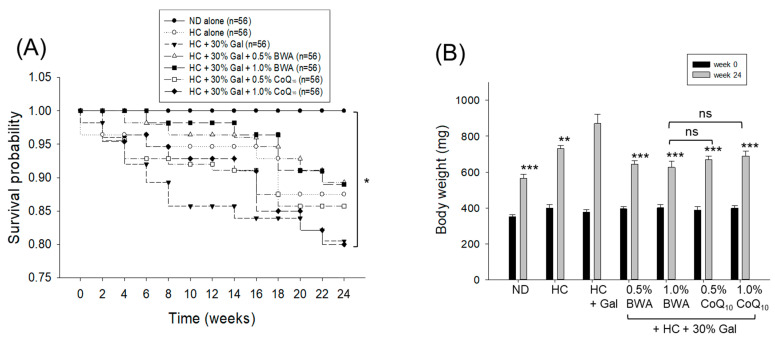
Zebrafish survivability and body weight across the different groups. (**A**) Survival probability across different groups during 24 weeks of consumption of the respective diets. The Kaplan–Meier survival analysis followed by log-rank test was used to determine the statistical difference in the survival probability curve. *, *p <* 0.05 (log-rank: *χ*^2^ = 13.5). (**B**) Body weight of zebrafish at the beginning (week 0) and after 24 weeks of consumption of the different diets. ND represents the normal diet; HC represents the high-cholesterol (final 4%, *w*/*w*) diet; HC+Gal represents the high-cholesterol (final 4%, *w*/*w*) with galactose (final 30%, *w*/*w*) diet; and HC+Gal+BWA (0.5%/1.0%) or CoQ_10_ (0.5%/1.0%) represents the high-cholesterol + galactose diet supplemented with beeswax alcohol (final 0.5% or 1.0% *w*/*w*) or coenzyme Q_10_ (final 0.5% or 1.0% *w*/*w*), respectively. ** (*p* < 0.01) and *** (*p* < 0.001) depict the statistical difference in the body weight measured at 24 weeks of consumption of different diets with respect to the HC+Gal group. “ns” represents the non-significant difference between the specified group concerning the HC+Gal+1.0% BWA group.

**Figure 3 antioxidants-13-01488-f003:**
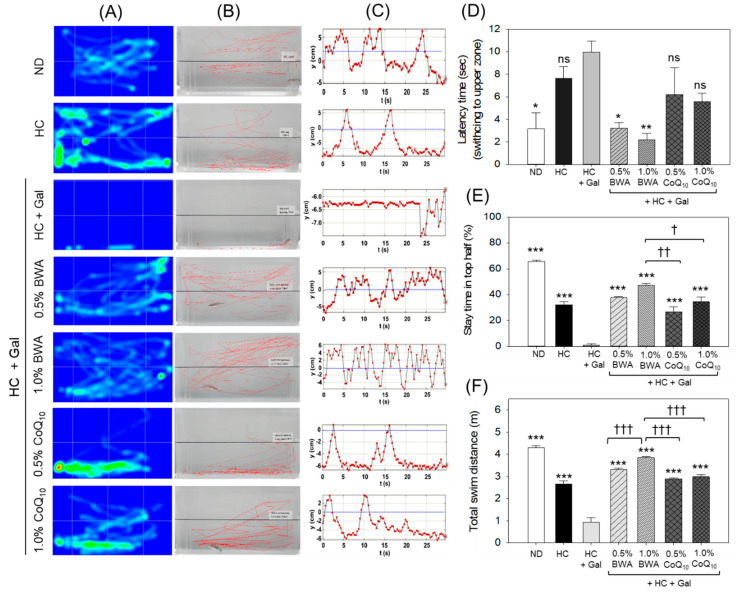
A representative image of the swimming trajectories and swimming behavior of zebrafish across different groups. (**A**) Heat map of zebrafish swimming trajectory for 60 s following 22 weeks consumption of different diets. (**B**) Swimming trajectory of zebrafish motion (60 s) across different groups after 24 weeks of consumption of different diets. (**C**) The vertical movement trajectories of zebrafish across different groups. The blue dotted line inside the images represents the middle line segregating the upper and lower halves of the water tank. (**D**) The latency time of movement (represents the average time zebrafish took to transition from the lower to the upper half) in 60 s. (**E**) Time of stay in the top half of the tank and (**F**) total swimming distance during 60 s. ND represents the normal diet; HC represents the high-cholesterol (final 4%, *w*/*w*) diet; HC+Gal represents the high-cholesterol (final 4%, *w*/*w*) with galactose (final 30%, *w*/*w*) diet; and HC+Gal+BWA (0.5%/1.0%) or CoQ_10_ (0.5%/1.0%) represents the high-cholesterol + galactose diet supplemented with beeswax alcohol (final 0.5% or 1.0% *w*/*w*) or coenzyme Q_10_ (final 0.5% or 1.0% *w*/*w*), respectively. * (*p* < 0.01), ** (*p* < 0.01) and *** (*p* < 0.001) depict the statistical difference with respect to the HC+Gal group, while ^†^ (*p* < 0.05), ^††^ (*p* < 0.01), and ^†††^ (*p* < 0.01) depict the statistical difference with respect to the HC+Gal+1.0% BWA group. “ns” represents the non-significant difference between the specified group concerning the HC+Gal group.

**Figure 4 antioxidants-13-01488-f004:**
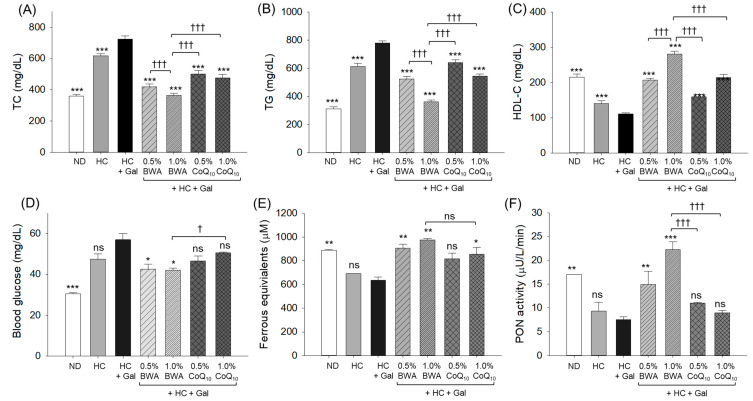
Blood lipid profile, blood glucose level, and antioxidant status of zebrafish (in different groups) consuming the specified diets for 24 weeks. (**A**) Total cholesterol (TC), (**B**) triglycerides (TG), (**C**) high-density lipoprotein cholesterol (HDL-C), (**D**) blood glucose level, (**E**) ferric ion reduction assay (FRA), and (**F**) paraoxonase (PON) activity. ND represents the normal diet; HC represents the high-cholesterol (final 4%, *w*/*w*) diet; HC+Gal represents the high-cholesterol (final 4%, *w*/*w*) with galactose (final 30%, *w*/*w*) diet; and HC+Gal+BWA (0.5%/1.0%) or CoQ_10_ (0.5%/1.0%) represents the high-cholesterol + galactose diet supplemented with beeswax alcohol (final 0.5% or 1.0% *w*/*w*) or coenzyme Q_10_ (final 0.5% or 1.0% *w*/*w*), respectively. * (*p* < 0.05), ** (*p* < 0.01) and *** (*p* < 0.001) depicts the statistical difference with respect to the HC+Gal group, while ^†^ (*p* < 0.001) and ^†††^ (*p* < 0.001) depicts the statistical difference with respect to the HC+Gal+1.0% BWA group. “ns” represents the non-significant difference between the specified group concerning the HC+Gal group or HC+Gal+1.0% BWA group.

**Figure 5 antioxidants-13-01488-f005:**
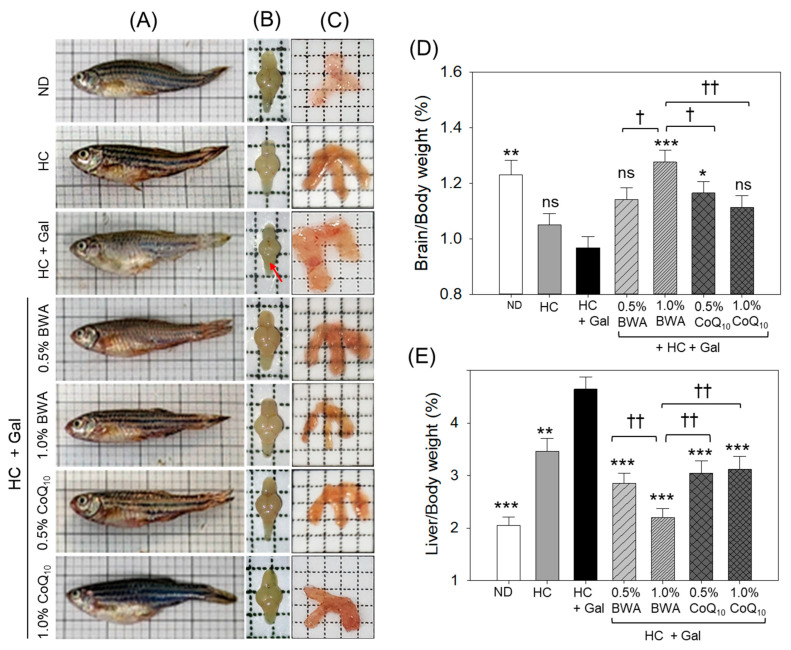
Pictorial view of representative morphological images of (**A**) zebrafish and (**B**) whole brain, with the red arrow depicting shrinkage of the cerebellum of the brain; (**C**) liver segregated from the different groups at 24 weeks of consumption of the designated diets. Percentage average weight (n = 15) of (**D**) brain and (**E**) liver with respect to body weight across the different groups consuming specified diets. ND represents the normal diet; HC represents the high-cholesterol (final 4%, *w*/*w*) diet; HC+Gal represents the high-cholesterol (final 4%, *w*/*w*) with galactose (final 30%, *w*/*w*) diet; and HC+Gal+BWA (0.5%/1.0%) or CoQ_10_ (0.5%/1.0%) represents the high-cholesterol + galactose diet supplemented with beeswax alcohol (final 0.5% or 1.0% *w*/*w*) or coenzyme Q_10_ (final 0.5% or 1.0% *w*/*w*), respectively. * (*p* < 0.01), ** (*p* < 0.01), and *** (*p* < 0.001) depict the statistical difference with respect to the HC+Gal group, while ^†^ (*p* < 0.05) and ^††^ (*p* < 0.01) depict the statistical difference with respect to the HC+Gal+1.0% BWA group. “ns” represents the non-significant difference between the specified group concerning the HC+Gal group.

**Figure 6 antioxidants-13-01488-f006:**
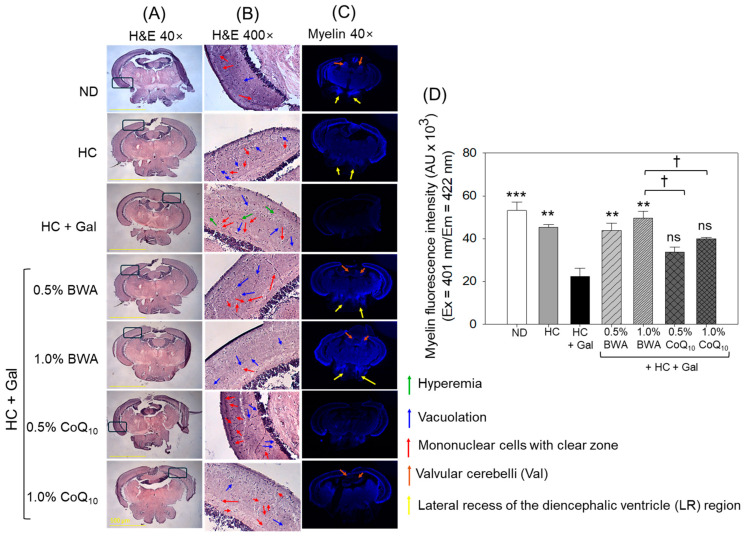
Brain histology of zebrafish at 24 weeks of consumption of the specified diets. (**A**) Images of hematoxylin and eosin (H&E) staining captured at 40× magnification, [scale bar, 0.5 mm]. (**B**) The digitally magnified images of the section inside the black box (as mentioned in (**A**)). (**C**) Immunohistochemical (IHC) images of myelin sheath in the brain section captured at 40× magnification. The yellow arrows highlight the myelin sheath in the lateral recess of the diencephalic ventricle (LR) region of the brain. (**D**) Image J software (version 1.53, https://imagej.net/ij; accessed on 16 June 2023) quantification of myelin fluorescent intensity. ND represents the normal diet; HC represents the high-cholesterol (final 4%, *w*/*w*) diet; HC+Gal represents the high-cholesterol (final 4%, *w*/*w*) with galactose (final 30%, *w*/*w*) diet; and HC+Gal+BWA (0.5%/1.0%) or CoQ_10_ (0.5%/1.0%) represents the high-cholesterol + galactose diet supplemented with beeswax alcohol (final 0.5% or 1.0% *w*/*w*) or coenzyme Q_10_ (final 0.5% or 1.0% *w*/*w*), respectively. ** (*p* < 0.01), and *** (*p* < 0.001) depict the statistical difference with respect to the HC+Gal group, while ^†^ (*p* < 0.05) depict the statistical difference with respect to the HC+Gal+1.0% BWA group. “ns” represents the non-significant difference between the specified group concerning the HC+Gal group.

**Figure 7 antioxidants-13-01488-f007:**
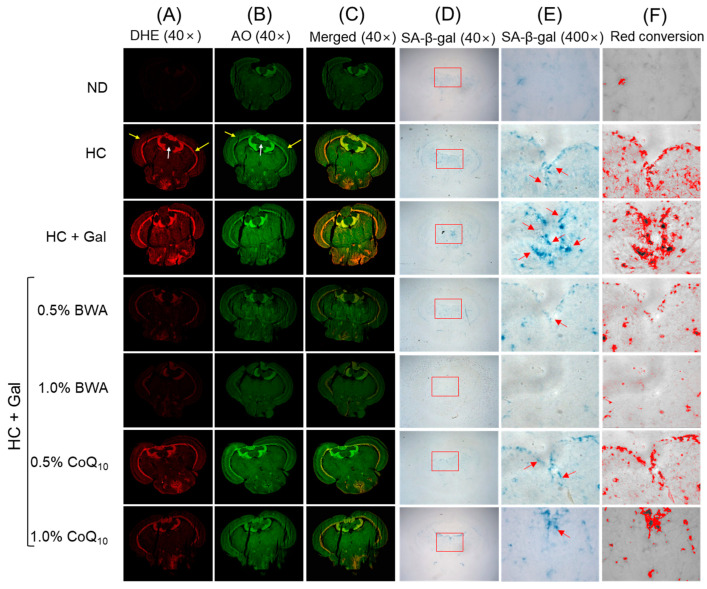
Fluorescent staining: (**A**) dihydroethidium (DHE), (**B**) acridine orange (AO), (**C**) merged images of DHE and AO captured at 40× magnification. (**D**) Cellular senescence-associated-β-galactose (SA-β-gal) staining at 40× magnification. (**E**) The 400× magnified images of the SA-β-gal-stained section inside the red box (as mentioned in (**D**)). The yellow arrow represents the periventricular gray zone of the optic tectum. The white arrow depicts the valvular cerebelli while the red arrow highlights the SA-β-gal positive cells around the vascular lacuna of the postrema and below the tectal ventricle region. (**F**) Red conversion of the (400× magnified) blue-colored SA-β-gal-positive cells. The red conversion was performed with Image J software (version 1.53) at the blue color threshold value (0–120) to enhance the visibility of the SA-β-gal-stained area. (**G**) and (**H**) Quantification of DHE and AO fluorescent intensity, respectively. (**I**) Quantification of the SA-β-gal-stained area. ND represents the normal diet; HC represents the high-cholesterol (final 4%, *w*/*w*) diet; HC+Gal represents the high-cholesterol (final 4%, *w*/*w*) with galactose (final 30%, *w*/*w*) diet; and HC+Gal+BWA (0.5%/1.0%) or CoQ_10_ (0.5%/1.0%) represents the high-cholesterol + galactose diet supplemented with beeswax alcohol (final 0.5% or 1.0% *w*/*w*) or coenzyme Q_10_ (final 0.5% or 1.0% *w*/*w*), respectively. ** (*p* < 0.01) and *** (*p* < 0.001) depict the statistical difference with respect to the HC+Gal group, while ^†^ (*p* < 0.05) and ^†††^ (*p* < 0.001) depict the statistical difference with respect to the HC+Gal+1.0% BWA group. “ns” represents the non-significant difference between the specified group concerning the HC+Gal group.

**Figure 8 antioxidants-13-01488-f008:**
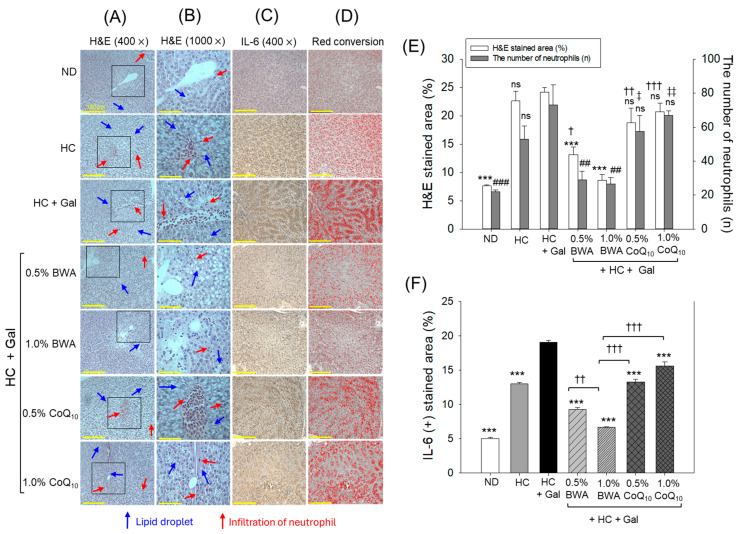
Hepatic histology of zebrafish across different groups consuming the specified diets for 24 weeks. (**A**) Hematoxylin and eosin (H&E) staining at 400× magnifications. (**B**) A 1000× magnified view of the H&E-stained section inside the black box (as mentioned in (**A**)). The red and blue arrows indicate neutrophil and lipid accumulation. (**C**) Interleukin (IL)-6-stained area examined by immunohistochemistry (IHC). (**D**) Brown-colored (IL-6 stained) area interchanged with red at the brown color threshold value 20–120 to enhance visibility. [Scale bar, 0.1 mm]. (**E**) Quantification of H&E-stained area and the number of neutrophils. (**F**) Percentage quantification of the IL-6-stained area. ND represents the normal diet; HC represents the high-cholesterol (final 4%, *w*/*w*) diet; HC+Gal represents the high-cholesterol (final 4%, *w*/*w*) with galactose (final 30%, *w*/*w*) diet; and HC+Gal+BWA (0.5%/1.0%) or CoQ_10_ (0.5%/1.0%) represents the high-cholesterol + galactose diet supplemented with beeswax alcohol (final 0.5% or 1.0% *w*/*w*) or coenzyme Q_10_ (final 0.5% or 1.0% *w*/*w*), respectively. *** (*p* < 0.001) (for H&E stained and IL-6-stained area), ^##^ (*p* < 0.01), and ^###^ (*p* < 0.001) (for neutrophil counts) depict the statistical difference with respect to the HC+Gal group. ^†^ (*p* < 0.05), ^††^ (*p* < 0.01), and ^†††^ (*p* < 0.001) (for H&E stained and IL-6-stained area), and ^‡^ (*p* < 0.05) and ^‡‡^ (*p* < 0.001) depict the statistical difference with respect to the HC+Gal+1.0% BWA group, respectively. “ns” represents the non-significant difference between the specified group concerning the HC+Gal group.

**Figure 9 antioxidants-13-01488-f009:**
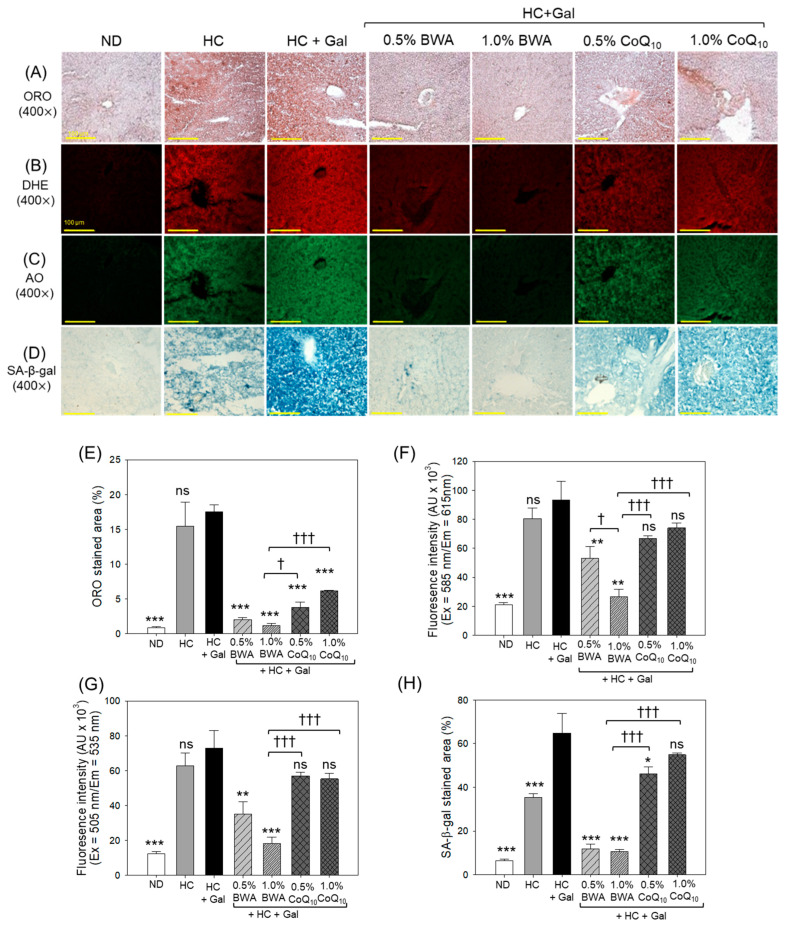
Evaluation of (**A**) oil red O (ORO) staining, (**B**,**C**) dihydroethidium (DHE) and acridine orange (AO) fluorescent staining, and (**D**) senescence associated-βgalactose (SA-β-gal) staining in hepatic tissue following 24 weeks of consumption of the specified diets. (**E**) Quantification of the ORO-stained area. (**F**) and (**G**) DHE and AO fluorescent intensity quantification, respectively. (**H**) Quantification of SA-β-gal-stained areas. ND represents the normal diet; HC represents the high-cholesterol (final 4%, *w*/*w*) diet; HC+Gal represents the high-cholesterol (final 4%, *w*/*w*) with galactose (final 30%, *w*/*w*) diet; and HC+Gal+BWA (0.5%/1.0%) or CoQ_10_ (0.5%/1.0%) represents the high-cholesterol+galactose diet supplemented with beeswax alcohol (final 0.5% or 1.0% *w*/*w*) or coenzyme Q_10_ (final 0.5% or 1.0% *w*/*w*), respectively. * (*p* < 0.05), ** (*p* < 0.01), and *** (*p* < 0.001) depict the statistical difference with respect to the HC+Gal group. ^†^ (*p* < 0.05) and ^†††^ (*p* < 0.001) depict the statistical difference with respect to the HC+Gal+1.0% BWA group, respectively. “ns” represents the non-significant difference between the specified group concerning the HC+Gal group.

**Figure 10 antioxidants-13-01488-f010:**
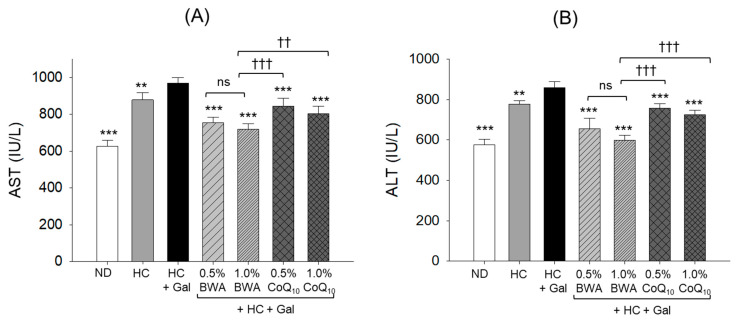
Plasma hepatic function biomarkers—(**A**) aspartate aminotransferase (AST) and (**B**) alanine aminotransferase (ALT)—in different groups consuming the specified diets for 24 weeks. ND represents the normal diet; HC represents the high-cholesterol (final 4%, *w*/*w*) diet; HC+Gal represents the high-cholesterol (final 4%, *w*/*w*) with galactose (final 30%, *w*/*w*) diet; and HC+Gal+BWA (0.5%/1.0%) or CoQ_10_ (0.5%/1.0%) represents the high-cholesterol + galactose diet supplemented with beeswax alcohol (final 0.5% or 1.0% *w*/*w*) or coenzyme Q_10_ (final 0.5% or 1.0% *w*/*w*), respectively. ** (*p* < 0.01) and *** (*p* < 0.001) depict the statistical difference with respect to the HC+Gal group, while ^††^ (*p* < 0.01) and ^†††^ (*p* < 0.001) depict the statistical difference with respect to the HC+Gal+1.0% BWA group. “ns” represents the non-significant difference between the specified group concerning the HC+Gal group.

**Figure 11 antioxidants-13-01488-f011:**
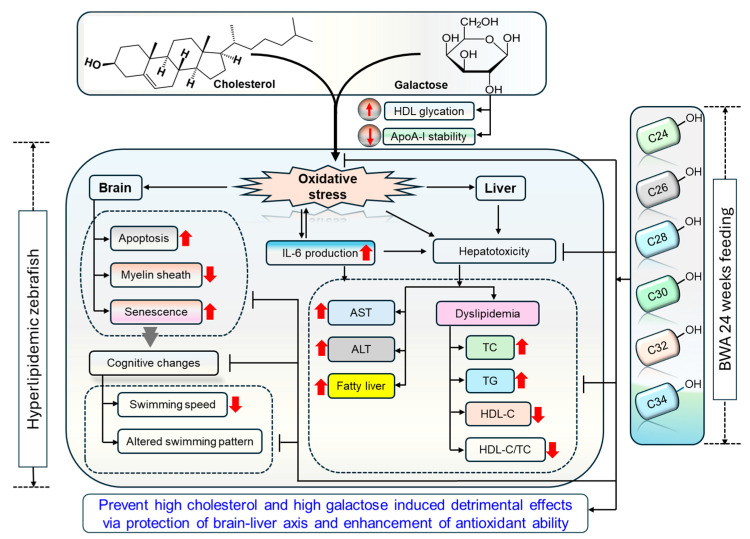
A summary of the protective effect exerted by beeswax alcohol (BWA) consumption to mitigate high-cholesterol + galactose (HC+Gal)-induced adversity in hyperlipidemic zebrafish.

**Table 1 antioxidants-13-01488-t001:** Seven distinct zebrafish diet compositions.

Diet (mg)	ND	HC	HC+30% Gal	HC+30% Gal+0.5% BWA	HC+30% Gal+1.0% BWA	HC+30% Gal+0.5% CoQ_10_	HC+30% Gal+1.0% CoQ_10_
Tetrabits ^1^	10	9.6	6.6	6.55	6.5	6.55	6.5
Galactose	0	0	3	3	3	3	3
Cholesterol	0	0.4	0.4	0.4	0.4	0.4	0.4
BWA	0	0	0	0.05	0.1	0	0
CoQ_10_	0	0	0	0	0	0.05	0.1
Total (mg)	10	10	10	10	10	10	10

ND, normal diet; HC, high cholesterol; BWA, beeswax alcohol; CoQ_10_, coenzymeQ_10_. ^1^ Tetrabits, a brand-name zebrafish diet, was purchased from Tetrabit Gmbh, Melle, Germany (containing 47.5% crude protein, 6.5% crude fat, 2.0% crude fiber, 10.5% crude ash, vitamin A (29,770 IU/kg), vitamin D3 (1860 IU/kg), vitamin E (200 mg/kg), and vitamin C (137 mg/kg)).

**Table 2 antioxidants-13-01488-t002:** Food consumption and food efficiency among the seven distinct zebrafish groups.

Dietary Efficiency	ND	HC	HC+30% Gal	HC+30% Gal+0.5% BWA	HC+30% Gal+1.0% BWA	HC+30% Gal+0.5% CoQ_10_	HC+30% Gal+1.0% CoQ_10_
Food consumption (%) ^1^	95–100	95–100	95–100	95–100	95–100	95–100	95–100
Food efficiency (%) ^2^	6.3	9.8	14.6	7.4	6.8	8.3	8.6

ND, normal diet; HC, high cholesterol; BWA, beeswax alcohol; CoQ_10_, coenzymeQ_10_. ^1^ The food consumption was calculated by [(given food amount − remaining food amount)/given food amount] × 100. ^2^ The feed efficiency was calculated at 24 weeks of consumption of the respective diets using [weight gain (g)/feed consumed (g)] × 100.

## Data Availability

The data used to support the findings of this study are available from the corresponding author upon reasonable request.

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
