# Peer review of "The Consumption of Beeswax Alcohol (BWA, Raydel®) Improved Zebrafish Motion and Swimming Endurance by Protecting the Brain and Liver from Oxidative Stress Induced by 24 Weeks of Supplementation with High-Cholesterol and D-Galactose Diets: A Comparative Analysis Between BWA and Coenzyme Q10"

_antioxidants, 2024, doi:10.3390/antiox13121488_

Round 1

Reviewer 1 Report

The authors have examined the protective effect of beeswax alcohol consumption, compared to that of coenzyme Q10, in a zebrafish model of high cholesterol (HC) and galactose (Gal) diet-induced oxidative stress and associated tissue damage. To that effect, they recorded changes in a number of behavioral, morphological, histological, and biochemical parameters, such as swimming behavior, plasma lipid profile, brain and liver morphology and histology, ROS production, apoptosis and senescence in brain and fatty liver tissue, and plasma parameters (aminotransferases) of liver damage, after 24 weeks of supplementation. Their work is an apparent extension of a previous study of shorter duration (12 weeks) (Cho KH et al. Pharmaceuticals 2024; 17(9): 1250), and their results were largely confirmatory, in that beeswax appears to exert a more protective effect. The two studies are not directly comparable however, as the composition of the HC-Gal diet differs considerably between the two (HC-Gal of 10%-10% in the previous study vs. 4%-30% in this one). I think the authors should comment on this difference, as well as on whether a 30% w/w Gal is a clinically relevant concentration. Also, I don’t quite see the relevance of the experiments evaluating the relative damage inflicted by the different hexoses, since the authors have already chosen to work with galactose in the past, and there is ample evidence in the scientific literature supporting that choice in this particular setup.

A few comments regarding presentation: i) please use Glc as an abbreviation for glucose, as Glu usually denotes glutamate, ii) in figure 3B, there is no indication of the statistical significance (if any), between the various treatments (BWA, CoQ10), iii) in all legends, “ns” appears to indicate lack of a statistically significant difference between any group and the reference group (HC+Gal), not “all groups”, iv) certain word uses are not very appropriate in their relevant context (e.g., “Contract to…”, “Conversely…” etc., in stead of “In contrast to…”, “As opposed to…”, “On the other hand…” etc.            

Author Response

Thank you for your insightful comments. Following the reviewer’s suggestion, we made point-to-point response and reflected on revision.

Please find attached doc as our response.

Reviewer 2 Report

This study compared the beneficial effects of dietary BWA and CoQ10 supplementations on the motion and swimming endurance of zebrefish offered the HC+Gal diet focusing on the brain and liver. The topic is interesting, and can attract broad interest. The novelty is also unquestioned. The manuscript was arranged in a straight-forward way, and is well written. There are several comments to heltp to further improve the quality of the manuscript.

1.      The absence of the high-Gal feeding group should be justified.

2.      Muscle is closely involved in the motion and swimming activity of fish. Why not choose it as the target tissue?

3.      Please justify the use of HDL3 derived from the human blood. Is it more reasonable to use the piscine HDL3?

4.      Although the levels of both ROS and IL-6 were measured, more parameters are needed to validate the anti-oxidant and anti-inflammatory effects of BWA and CoQ10 in zebrafish.

5.      Feed consumption and efficiency, plasma glucose levels as well as tissue glycogen contents should be measured.

6.      Concerning Table 1, the part of diet composition should be moved into the M & M section. In addition, the survivability and BW data are partially repetitive with those presented in Fig. 3, and are recommended to be deleted. 

7.      Data presented in both Fig. 2A and 3A were recommended to be analyzed by two-way ANOVA.

8.      The Results section is quite long and redundant, and needs to be shortened.

9.      The Discussion section should be re-organized to interpret the data from the perspective of the liver-brain axis.

10.  Others: 1) Please rephrase the sentence in lines 56-59 to fit into the introduction section; 2) The taxonomic name of an organism should be provided when firstly mentioned, like zebrafish in line 57.

Author Response

(The authors gave the same response as above.)

Round 2

Reviewer 2 Report

This study compared the beneficial effects of dietary BWA and CoQ10 supplementations on the motion and swimming endurance of zebrefish offered the HC+Gal diet focusing on the brain and liver. The topic is interesting, and can attract broad interest. The novelty is also unquestioned. 

The authors have made substantial efforts to revise the manuscript. All the comments were addressed appropriately. There is only one minor issue left. The feed consumption and efficiency data should be presented in Table or Figure in the Results section. 

Author Response

Thank you for your insightful comments. Following the reviewer’s suggestion, we made point-to-point response and reflected on revision.

Please find attached doc as our response.

Thank you for your appreciation. The consumption efficiency data has been incorporated in Figure 2 of the revised manuscript. Kindly refer to the figure below and the corresponding section in the revised manuscript (Lines 261 & 264 and Table 2). Thank you!
